# Proteome-wide determinants of co-translational chaperone binding in bacteria

Carla Verónica Galmozzi[1,2,3,8], Frank Tippmann[1,8], Florian Wruck[4], Josef Johannes Auburger[1], Ilia Kats [1,5], Manuel Guennigmann [1], Katharina Till[4], Edward P. O'Brien [6], Sander J. Tans[4,7] ✉, Günter Kramer [1] ✉ & Bernd Bukau [1] ✉

Chaperones are essential to the co-translational folding of most proteins. However, the principles of co-translational chaperone interaction throughout the proteome are poorly understood, as current methods are restricted to few substrates and cannot capture nascent protein folding or chaperone binding sites, precluding a comprehensive understanding of productive and erroneous protein biosynthesis. Here, by integrating genome-wide selective ribosome profiling, single-molecule tools, and computational predictions using Alpha-Fold we show that the binding of the main *E. coli* chaperones involved in co-translational folding, Trigger Factor (TF) and DnaK correlates with "unsatisfied residues" exposed on nascent partial folds – residues that have begun to form tertiary structure but cannot yet form all native contacts due to ongoing translation. This general principle allows us to predict their co-translational binding across the proteome based on sequence only, which we verify experimentally. The results show that TF and DnaK stably bind partially folded rather than unfolded conformers. They also indicate a synergistic action of TF guiding intra-domain folding and DnaK preventing premature inter-domain contacts, and reveal robustness in the larger chaperone network (TF, DnaK, GroEL). Given the complexity of translation, folding, and chaperone functions, our predictions based on general chaperone binding rules indicate an unexpected underlying simplicity.

Proteins are particularly susceptible to misfolding and aggregation when they emerge from the ribosomal tunnel and begin to form structure[1]. Guidance by molecular chaperones is critical to this co-translational folding process across all domains of life, yet is poorly understood[2,3]. While chaperones interact generally with a vast spectrum of nascent proteins, whose size and structure vary radically during translation and throughout the proteome, these interactions must also be specific to affect folding productively[4]. This general-yet-specific binding mode is key to co-translational chaperone function and hints at universal underlying rules, but is difficult to address experimentally.

Best-characterized is *Escherichia coli*, where the chaperone trigger factor (TF) directly binds ribosomes near the exit tunnel and interacts with a large fraction of emerging nascent chains[5]. The DnaK system[6,7]

---

[1]Center for Molecular Biology of Heidelberg University (ZMBH), DKFZ-ZMBH Alliance, Heidelberg, Germany. [2]Instituto de Biomedicina de Sevilla, Hospital Universitario Virgen del Rocío/CSIC/Universidad de Sevilla, Seville, Spain. [3]Departamento de Genética, Facultad de Biología, Universidad de Sevilla, Seville, Spain. [4]AMOLF, Amsterdam, The Netherlands. [5]Division of Computational Genomics and Systems Genetics, German Cancer Research Center (DKFZ), Heidelberg, Germany. [6]Department of Chemistry, Pennsylvania State University, University Park, Pennsylvania, PA, USA. [7]Department of Bionanoscience, Kavli Institute of Nanoscience Delft, Delft University of Technology, Delft, The Netherlands. [8]These authors contributed equally: Carla Verónica Galmozzi, Frank Tippmann. ✉e-mail: tans@amolf.nl; g.kramer@zmbh.uni-heidelberg.de; bukau@zmbh.uni-heidelberg.de

also engages many proteins co-translationally[8–11], while the range is less clear for other chaperones like GroEL-ES[12–14]. Knocking out either TF or DnaK affects growth only modestly. However, a dual knock-out is synthetically lethal above 30 °C and causes widespread protein aggregation[15,16]. Both TF and DnaK are thought to bind unfolded nascent chain conformers, with the cradle-shaped TF engaging in multivalent low-affinity contacts of hydrophobic and polar origin[17–20], and DnaK using a binding groove and nucleotide-controlled lid to capture short peptide segments enriched in hydrophobic residues flanked by basic residues[6,7]. In this way, TF and DnaK are proposed to shield nascent chains from aggregation[20–27], promote co-translational folding[26], as well as delay folding until later stages of translation[22,28]. Whether these modes of action can explain general and specific binding across the proteome is unclear. Here, we aimed to elucidate the determinants of TF and DnaK co-translational binding throughout the proteome, and to study their interplay within the TF, DnaK, and GroEL-ES network. Structural methods like NMR and Cryo-EM can show how nascent chains and chaperones interact[29], while biochemical and single-molecule techniques can probe folding[30,31]. However, these approaches necessarily study certain substrates at certain moments of translation, limiting proteome-wide views.

We integrated three methods to overcome these experimental limitations. Selective ribosome profiling[32,33] (SeRP) showed when TF, DnaK, and GroEL bind during translation across the proteome. Optical tweezers[34] experiments combined with fluorescence[35] showed that TF and DnaK bind partially folded conformers more stably than unfolded ones, leading us to hypothesize that they recognize "unsatisfied residues"–residues that cannot yet form all native contacts due to incomplete translation–that are exposed on nascent folded structures. Our data suggest that TF and DnaK play distinct roles in co-translational folding, with TF chaperoning intra-domain folding and DnaK preventing premature inter-domain contacts. Furthermore, we used SeRP to study the proteome-wide response of one chaperone upon the deletion of another. These data revealed that the chaperone network uses a combinatorial program to assist in adaptive folding of the nascent chain proteome. We identified unsatisfied residues exposed on partially folded structures as important proteome-wide determinants of co-translational chaperone guidance in *E. coli*.

## Results

### Co-translational engagement of TF, DnaK, and GroEL throughout the proteome

We performed SeRP to comprehensively study binding of the major cytosolic chaperones TF, DnaK, and GroEL to nascent chains. SeRP is based on two deep sequencing datasets (Fig. 1a). The total translatome is defined by all the (approximately) 30-nucleotide-long mRNA footprints that are protected by ribosomes at cell harvest. In contrast, the chaperone-selected translatome enriches the mRNA footprints of ribosomes whose nascent chains are engaged by one of the chaperones, as obtained by affinity purification (AP) for that chaperone before sequencing. Finally, the selected translatome is normalized by the total translatome in a codon-wise manner, thus providing the chaperone enrichment profiles along all open reading frames (ORFs) (Fig. 1a, Supplementary Fig. 1a, b, j). Metagene profiles (averaged over all ORFs) show that TF preferentially binds at chain lengths of 100–250 residues (Fig. 1b, Supplementary Fig. 1c). DnaK and GroEL engagement instead increases continuously during translation, suggesting a general preference for long nascent chains and consecutive chaperone action. We determined the maximum enrichment for each ORF (chaperone engagement scores[36]) to rank proteins according to their strength of interaction (Supplementary Fig. 1e left, Supplementary Data 1). The engagement scores differ strongly between chaperones, but they mainly indicate differences in substrate pool size (TF>DnaK >> GroEL), as ubiquitously acting chaperones are

distributed among more substrates, which leads to smaller enrichment values.

Next, we aimed to rank protein substrates for each chaperone separately. Engagement scores are broadly distributed, making a threshold-based selection of substrate groups unsuitable (Supplementary Fig. 1d). Thus, we performed pair-wise comparisons of chaperone engagement profiles using dynamic time warping, a method to compare the variation between two signals of different lengths[37], and employed the resulting matrix for hierarchical clustering (Supplementary Fig. 1e). For each chaperone, this revealed a high-affinity cluster A and a lower-affinity cluster B (Supplementary Fig. 1f, Supplementary Data 2). TF and DnaK each had more than 500 proteins within cluster A. Cluster A of GroEL instead comprises only 50 proteins, indicating GroEL has a much smaller nascent substrate pool, consistent with our engagement score ranking analysis. Within the clusters A, proteins with high sequence similarity also have highly similar chaperone enrichment profiles, suggesting that homologs have similar folding paths and chaperone assistance needs (Supplementary Fig. 1g). This is exemplified by similar DnaK engagement profiles of the homologous proteins LeuS and ValS (Supplementary Fig. 1f right and h), and the presence of specific protein classes in clusters A, like aminoacyl tRNA synthetases for TF and DnaK, histidine kinases for DnaK, and peroxidases for GroEL (Supplementary Data 3).

### Chaperone networking

To study how the individual chaperones integrate into a functional network, we first determined which nascent proteins are bound by none, one, or multiple chaperones (Fig. 1c). While TF and DnaK bind many nascent chains exclusively (Supplementary Fig. 1i left and mid), GroEL engagement is almost always accompanied by TF and/or DnaK, with only three detectable exceptions (e.g., Supplementary Fig. 1i right). All chaperones prefer binding larger proteins, which is most clearly visible for cluster A substrates of DnaK, and 21% of cluster A proteins that interact strongly with two or three chaperones (Fig. 1c, d). This suggests that larger proteins have a higher and more complex demand for co-translational folding assistance. Focusing on these proteins, we generated overlapping heatmaps of chaperone engagement that show how the chaperones cooperate on individual nascent chains (Fig. 1e). Implying exclusive chaperone action, TF binding periods during translation generally precede and hardly overlap with binding periods of DnaK and GroEL, and TF dissociation is often closely followed by binding of DnaK and/or GroEL. In contrast, DnaK and GroEL binding periods almost always overlap, suggesting similar binding preferences and time points of action during translation. Overlapping binding periods are compatible with either cooperative or exclusive, competitive binding of DnaK and GroEL to nascent chains. Overall, within this subgroup of multi-chaperone binders, a continuous and ordered program of chaperone engagement throughout translation is a frequent phenomenon (Fig. 1e).

### Emergence of folding domains promotes TF and DnaK binding

Using the information of TF and DnaK interaction with hundreds of high-affinity substrates, we investigated which nascent chain features promote chaperone binding. We explored whether TF binding to cytosolic substrates correlates with the exposure of sequence motifs during translation, inspired by the suggested preference of TF for hydrophobic stretches or peptides[7,30]. However, within the 150 residues that are emerged from the ribosomal exit tunnel at the onset of TF engagement, we did not detect (i) enhanced hydrophobicity, (ii) altered isoelectric points, (iii) accumulation of specific residues, or (iv) enrichment of secondary structure elements (Supplementary Fig. 2a). Similarly, we did not find any evidence that DnaK engages emerging chains exposing hydrophobic segments or specific residues (Supplementary Fig. 2b).

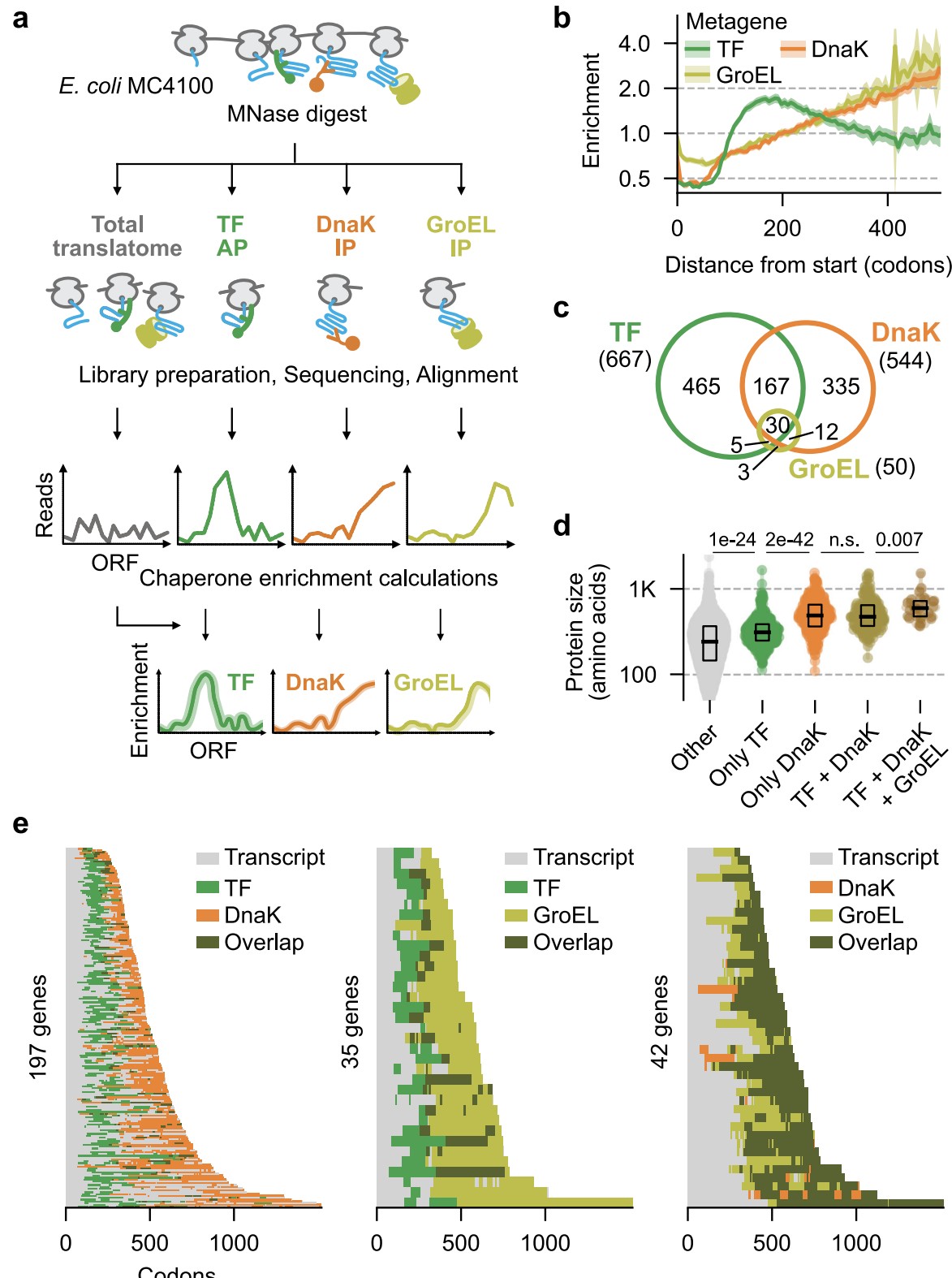

**Fig. 1 | Profiling of chaperone binding to nascent chains. a** Schematic of the SeRP experiments. AP affinity purification, IP immunoprecipitation, ORF open reading frame. **b** Genome-wide (*n* = 3961) metagene enrichment profiles for TF (green), DnaK (orange), and GroEL (olive). Solid lines and shadows indicate means and 95% CI, respectively. **c** Overlap between the strongest TF, DnaK, and GroEL interactors. Numbers indicate involved proteins. See Supplementary Fig. 1e, f for details.

**d** Protein size distribution of chaperone interactors (*n* = Only TF: 465; Only DnaK: 335; TF + DnaK: 167; TF + DnaK + GroEL: 30) and all other proteins (*n* = 2951). Black bars indicate medians, boxes indicate the interquartile range. *P* values were calculated using two-sided Mann–Whitney *U* tests. **e** Heat maps of chaperone binding periods for nascent chains of high-affinity substrates. Source data are provided as a Source Data file.

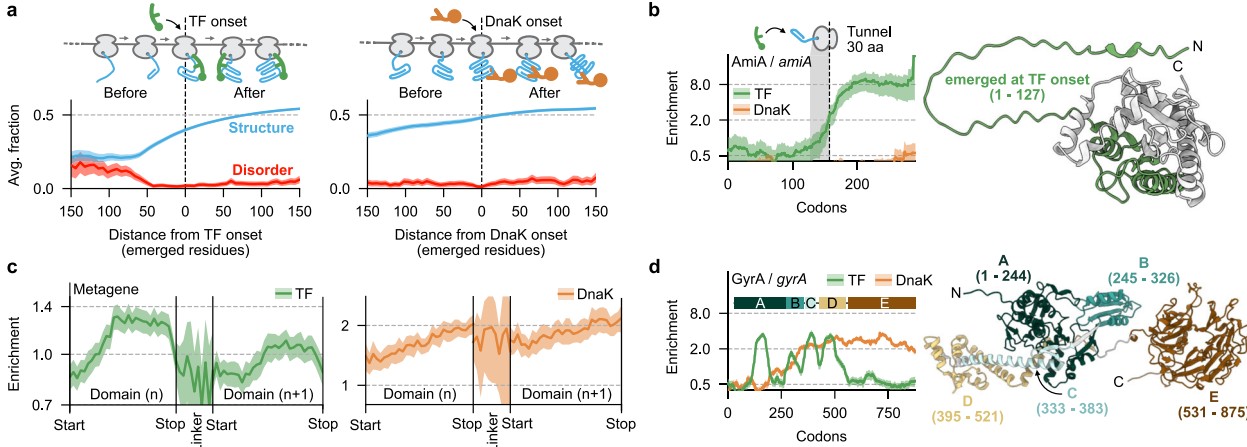

**Fig. 2 | Formation of protein domains determines TF binding. a** Average fraction (solid lines) of nascent chain residues with predicted disorder based on IUPred3-short[88] (red) or tertiary structure based on ribosomal tunnel-emerged residue-residue contacts in AlphaFold prediction (blue) for TF (left, $n = 429$) and DnaK (right, $n = 353$). Only cytosolic, high-affinity substrates were included. Curves are aligned to the engagement onset of chaperones (dashed lined) and shifted by 30 amino acids to the 5' end of transcripts to account for the ribosomal exit tunnel. Shadows indicate 95% CI. **b** TF (green) and DnaK (orange) enrichment profiles to nascent AmiA. The dashed line indicates the onset of TF interaction, and the grey area represents the ribosomal exit tunnel. Solid lines indicate averages, shadows indicate the 95% CI (left). AlphaFold prediction of the AmiA structure (AF-P36548). N-terminal residues (1–127) of AmiA that are ribosome-exposed at the onset of TF

engagement are colored in green (right). **c** Metagene enrichment profile of TF (green, left) and DnaK (orange, right) aligned to the start and stop positions of 2019 consecutive domain-pairs from 1014 cytosolic multi-domain transcripts. Domain and linker regions were summed in 30 and 10 bins, respectively. Linker regions longer than 40 amino acids are excluded. TF and DnaK engagement positions are shifted by 30 amino acids to the 5' end of transcripts to account for the ribosomal exit tunnel. Solid lines and shadows indicate means and 95% CI, respectively. **d** TF and DnaK enrichment profiles to nascent multi-domain protein GyrA (left). Colored bars above show the tunnel-normalized CATH domain annotation. Solid green and red lines indicate averages, shadows indicate the 95% CI. AlphaFold prediction of the GyrA structure (AF-P0AES4) with colored domain annotation, numbers show amino acid positions (right). Source data are provided as a Source Data file.

As primary and secondary structure features are not the major determinants for chaperone binding, we next explored the relevance of tertiary structure. We detected a decrease in predicted nascent chain disorder and an increase in tertiary structure associated with TF engagement (Fig. 2a left). Nascent chains might acquire a tertiary fold already before the onset of TF binding, which is exemplified by nascent AmiA. Here, TF does not bind the unstructured N-terminus but engages sharply after enough residues emerge to allow tertiary structure formation (Fig. 2b). Further, meta- and gene-specific enrichment profiles showed that TF binding begins upon emergence of CATH annotated domains and ends when these are fully exposed (Fig. 2c left). As observed for GyrA (Fig. 2d), the correlation of TF binding and domain emergence is most clearly visible for N-terminal domains and gradually fades out when nascent chains grow longer, in agreement with the TF preference for shorter nascent chains (Supplementary Fig. 2c).

To obtain independent evidence for the correlation between TF binding and nascent chain folding, we used limited proteolysis to analyze the folding state of nascent TrpD, a TF substrate. We purified ribosomes stalled when 172 TrpD residues had emerged from the exit tunnel, using the SecM stalling sequence. TF engagement then peaks, while the first TrpD domain is not fully exposed (Supplementary Fig. 2e, f). Subsequent digestion with increasing amounts of trypsin, followed by α-SecM immunoblotting, revealed an accumulation of two incompletely digested TrpD-SecM fragments larger than 10 kDa, thus protecting several trypsin cleavage sites. These data showed that the TrpD[172] fragment is indeed partially folded and contains tertiary structure when TF engages. Consistently, when we performed experiments on stalled ribosomes where the first TrpD domain was fully exposed (TrpD[192]), the nascent protein was considerably more resistant to trypsin, in line with complete folding of domain A. These results agree with our model that initial nascent chain tertiary structure formation promotes TF binding, while folding completion of domains drives TF dissociation.

In comparison, the DnaK metagene profile showed a different, less clear correlation with structure and domains (Fig. 2a, c right). DnaK enrichment instead gradually increases during domain emergence, but does not decrease upon single domain completion like for TF (Supplementary Fig. 2d), as also observed for nascent GyrA (Fig. 2d). Note that these gradual increases result from averaging between different substrates: profiles of individual substrates do show sharper increases in the DnaK engagement profiles (Fig. 2d, Supplementary Fig. 1h, i), which occur at different moments of translation and hence are smoothened to yield the gradually increasing metagene profile (Fig. 2c, right).

## TF and DnaK stably bind partially folded nascent proteins
To test more directly whether TF as well as DnaK bind nascent chains with tertiary structure, we developed an in vitro single-molecule assay (Fig. 3a, Supplementary Fig. 3a). We manipulated individual stalled ribosome nascent chain complexes (RNCs) by optical tweezers as shown before[38,39], which allowed us to study the same nascent chain in both the unfolded state and partially folded states (Fig. 3a, Supplementary Fig. 3b). We integrated simultaneous imaging of individual TF and DnaK molecules using two-color confocal fluorescence microscopy, as done previously for single proteins without ribosomes[35,40] (Fig. 3a, b, Supplementary Fig. 3c). DnaK and TF concentrations were maximized to detect binding, but could not exceed 100 nM due to the general problem of background fluorescence, a concentration which is significantly lower than observed in vivo. Hence, we focused on trends rather than absolute quantification. To manipulate single RNCs, we first tethered single ribosomes to micron-sized polystyrene beads via a DNA handle, then performed in vitro transcription-translation to generate stalled nascent chains, and subsequently attached a second bead to the nascent chain N-terminus using biotin-neutravidin linkages and another DNA handle (Fig. 3a, Supplementary Fig. 3a).

We studied AceE as it is among the top interactors for both TF and DnaK (Supplementary Data 1). We first investigated chaperone binding

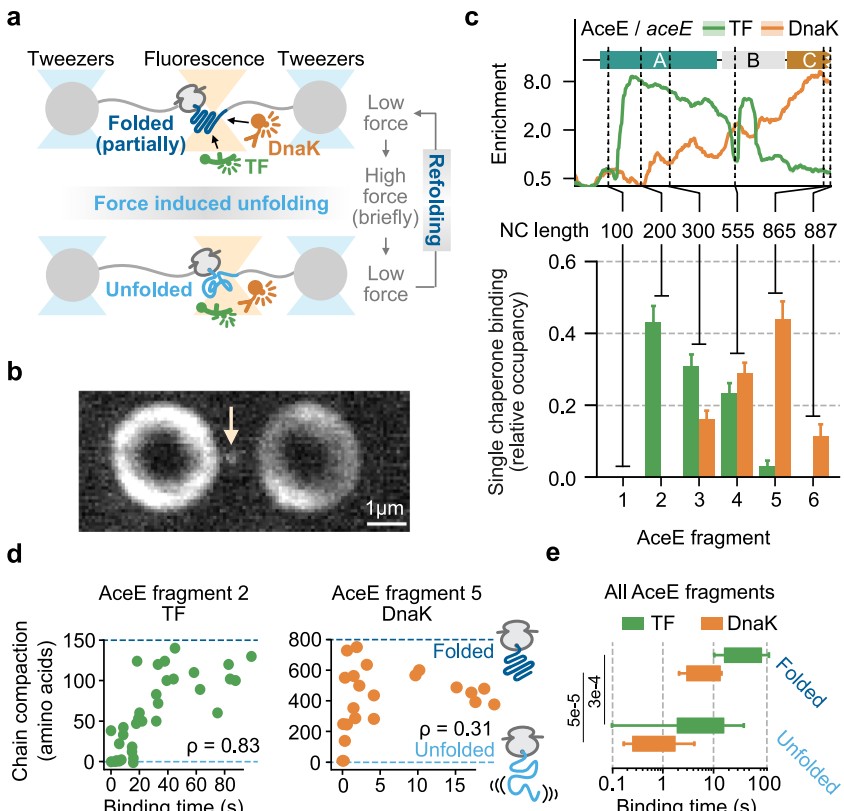

**Fig. 3 | Single-molecule characterization of chaperone binding and nascent chain folding state. a** Optical tweezer experimental setup. **b** Example confocal image of fluorescently labeled TF immobilized on a tethered ribosome nascent chain complex (arrow). In total 173 TF binding events were measured (see methods for details). **c** AceE enrichment profiles from SeRP. Solid green and red lines indicate averages, and shadows indicate 95% CI (top). TF and DnaK binding statistics for 6 AceE fragments (173 TF binding events and 162 DnaK binding events). Bars and lines show the fractional frequency and standard error of proportion (bottom, see methods for details). **d** Measured compaction states versus TF binding times for AceE fragment 2 ($n = 33$, left) and DnaK binding times for AceE fragment 5 ($n = 21$, right). $\rho$ indicates the Spearman correlation coefficient. **e** Binding time of TF and DnaK to AceE nascent chains that are predominantly folded (top) or predominantly unfolded (bottom). $n$: TF, folded: 21, TF unfolded: 19, DnaK, folded: 12, DnaK unfolded: 21. Boxes indicate the interquartile range, whiskers are drawn down to the 10th and 90th percentile. $P$ values were calculated using two-sided Mann–Whitney $U$ tests. Source data are provided as a Source Data file.

dependence on nascent chain length (Fig. 3c top). We stalled RNCs at 6 sites along the *aceE* mRNA, and quantified chaperone engagement for each, using multiple tethered RNCs and chaperone binding events (Fig. 3c bottom). TF engagement was undetectable for the AceE fragment of 100 residues, peaked at 200 residues, and then gradually decreased to zero at the end of the transcript. DnaK engagement was also undetectable for the shortest fragments, and increased for longer lengths (as TF engagement decreased), after finally decreasing again. These trends are consistent with those observed by SeRP in vivo (Fig. 3c, top). We did not detect simultaneous binding of TF and DnaK, consistent with our data suggesting the exclusive engagement of nascent chains by one chaperone (Fig. 1e).

The binding kinetics showed notable differences between TF and DnaK. Specifically, the time that DnaK remained bound to nascent AceE was about 10-fold shorter than for TF (Supplementary Fig. 3d). When the entire chaperone machinery was present (DnaK, DnaJ, GrpE), DnaK binding periods on nascent AceE were too short for proper quantification. Next, we analyzed how nascent tertiary structure impacts chaperone binding, first focusing on fragments that showed the most prominent binding (fragment 2 for TF, fragment 5 for DnaK). As one nascent chain is stretched and relaxed, it adopts by chance different stably folded states, which range from fully folded to fully unfolded (Fig. 3d, Supplementary Fig. 3b). Hence, we compared nascent chains of the same length but different levels of tertiary structure formation. TF was found to bind unfolded states briefly (order of seconds) while the binding times increased to over 50 seconds for

more folded nascent chains (Fig. 3d left). DnaK showed a similar though less strong effect. Nascent chains that were less than one-third folded showed residence times of DnaK below 5 seconds, while those that were folded by more than one-third showed residence times of 10 seconds and higher (Fig. 3d right). We performed such experiment for all fragments and found again similar trends: both TF and DnaK were bound on average 10-fold longer on predominantly folded nascent chains compared to predominantly unfolded chains, while overall DnaK bound less long than TF (Fig. 3e). Together, our data show that both TF and DnaK bind more stably to nascent AceE chains that have adopted a (partially) folded state.

## Proteome-wide prediction of chaperone binding

To study the determinants of chaperone binding across the cytosolic proteome, in particular the notion that TF and DnaK bind most stably to partial nascent folds, we used an in silico chaperone binding model. To develop this model, we did not use protein-specific information (e.g., from the SeRP data), except for the amino acid sequences of every cytosolic *E. coli* protein. The model takes the sequence and uses AlphaFold[41,42] structure prediction and general chaperone binding rules to predict TF and DnaK binding profiles for specific proteins, which in turn are compared to the SeRP data. In this way, we aimed to study whether these general chaperone binding rules have predictive value for specific proteins across the proteome.

We first focused on TF, for which we considered two general binding rules: (1) TF stably binds partial nascent folds, and (2) on these

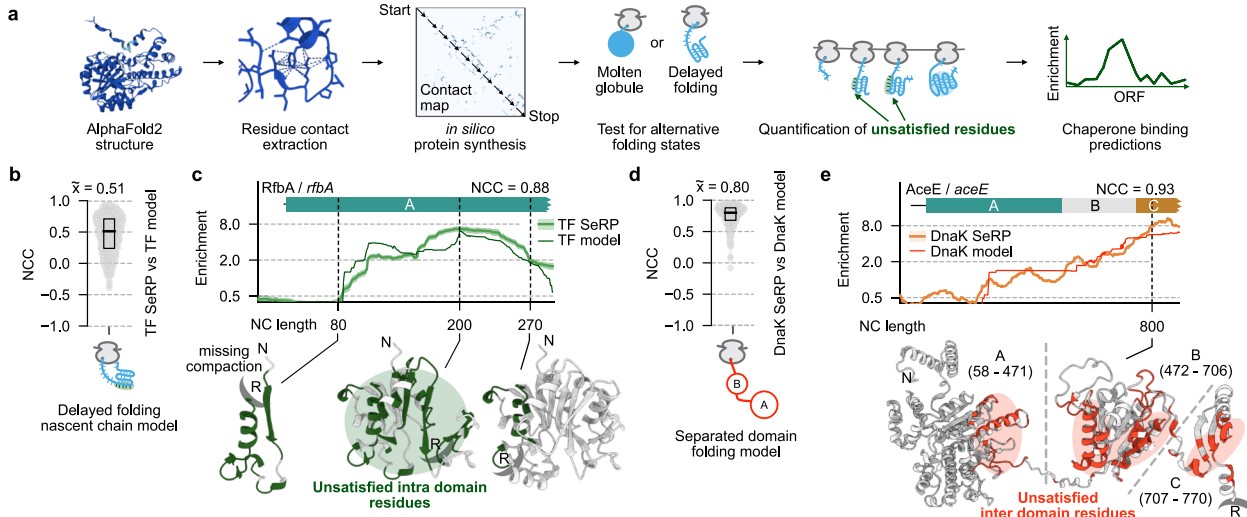

**Fig. 4 | Computational prediction of chaperone binding to nascent chains.**
**a** Schematic of the prediction layout. **b** Normalized cross correlation at tau = 0 (NCC) comparing TF model predictions with the SeRP derived low CI boundaries from strong cytosolic TF substrates (n = 443). Black bar and top number (x̃) indicate the median NCC while the box indicates the interquartile range. **c** TF model predictions (dark green) and SeRP-based TF enrichment profile (green) for RfbA. Solid thick line indicates the average, shadow indicates the 95% CI. For codons 80, 200, and 270 the tunnel-emerged regions of the AlphaFold predicted RfbA structure (AF-P37744) are shown. Unsatisfied intra-domain residues are shown in dark green. The ribosome surface is indicated by crescent shape and "R". **d** NCC comparing DnaK model predictions with the SeRP derived low CI boundaries from strong cytosolic DnaK substrates (n = 252) as in Fig. 4b. **e** Same as Fig. 4c but for AceE using DnaK SeRP-based data (orange) and the DnaK model (dark red). The AlphaFold prediction (AF-P0AFG8) at NC length 800 was modified to show domain separation. Unsatisfied inter-domain contacts are shown in dark red. Source data are provided as a Source Data file.

partial folds, TF engages sites that later become inaccessible when translation and folding of a domain are completed. To predict the strength of TF engagement, we combined both criteria into the notion of "unsatisfied residues": amino acids that cannot form all native contacts in a certain phase of translation, because partner residues have not yet emerged. As translation and folding are completed, these residues thus become satisfied, prompting TF to disengage. This also excludes unfolded residues that remain disordered in the native structure. An intuitive rationale is that multiple unsatisfied residues come together in a stable configuration on such tertiary structures and hence can be bound more stably by TF than the dynamic and transient interactions afforded by unfolded chains. Briefly, we used native structure predictions of all cytosolic *E. coli* proteins, then determined the residue-residue contacts within domains, and for every nascent chain length, determined the unsatisfied residues exposed on partial folds, which in turn defined the predicted TF binding scores (Fig. 4a, see methods for details). We thus assumed that nascent chains can adopt native-like contacts during translation, as suggested by numerous studies[43–49], except for the 30 residues residing in the ribosome tunnel. Given that entropy and kinetic factors[50–52] as well as ribosome proximity[22,30,53–57] can delay folding, we delayed the formation of partial native structures until a minimal number of native contacts are possible (see methods for details). Hence, the model only used protein sequences and general chaperone binding rules to predict specific TF binding profiles for each cytosolic protein.

For a large fraction of strongly binding cytosolic proteins (termed cluster A), this "TF model" predicted the SeRP binding profiles well (median cross-correlation value of 0.51, Fig. 4b, Supplementary Fig. 4a). An example is nascent RfbA, which starts engaging TF when the first 80 residues are translated, both in the SeRP data and the model (Fig. 4c). The model also correctly predicts that TF binding increases up to codon 200, caused by newly emerged residues that guide the formation of additional partial folds. Subsequently translated residues instead allow complete folding of the domain, thus causing TF dissociation, in agreement with the SeRP data (Fig. 4c). The model also accurately predicts TF binding to many multi-domain proteins (e.g., RibD or PurF, Supplementary Fig. 4b). For some nascent

proteins the onset and/or efficiency of TF binding were predicted less well, though notably the number of binding peaks did match (PheT and TldD, Supplementary Fig. 4c). One likely source of error is the complexity of the folding process for the large proteins considered here, which is challenging to fully capture.

We also tested an alternative model where TF instead binds to a dynamic molten globule state[58] of nascent chains. TF then dissociates when domain synthesis is completed and folding takes place. This "molten globule" model poorly predicted the TF SeRP-binding profiles, except for a few small single-domain proteins (Supplementary Fig. 4d, e). Allowing early native contacts between secondary structure or hydrophobic (I, L, V, F) residues in the molten globule did not improve this model (Supplementary Fig. 4f).

The DnaK model we considered is similar to the TF model, with one key difference: DnaK primarily binds those unsatisfied residues on partially folded structures that are involved in inter-domain contacts, whereas TF binds those involved in intra-domain contacts. This was inspired by the experimentally observed increased DnaK enrichment during domain emergence (Fig. 2c right, d) and longer binding to (partially) folded structures (Fig. 3d, e). Note that, like for TF, this model does not contain or use information specific to individual proteins from the SeRP data. We assume that the native inter-domain contacts form post-translationally, and thus remain accessible for DnaK during translation, following the Pearl-Necklace-Like theory[59,60] (Fig. 4d, bottom). The model correctly predicted DnaK binding to a large number of nascent chains (median cross-correlation value of 0.8). One example is nascent AceE, for which the model correctly predicts the DnaK engagement onset when multiple unsatisfied inter-domain residues are brought together by folding (Fig. 4e). Other examples are the increasing DnaK binding to nascent LepA and MetH that follows the growing number of unsatisfied interface residues during translation (Supplementary Fig. 4g).

Metagene SeRP profiles of all cytosolic proteins for TF and DnaK were also well predicted by our models (Supplementary Fig. 4h). In summary, the observed chaperone binding to partial folds and the resulting concept of massed unsatisfied residues enabled prediction of co-translational TF and DnaK binding throughout the cytosolic cluster A substrates (Supplementary Fig. 4i).

## Robustness of the chaperone network

The growth defects caused by combined TF (Δ*tig*) and DnaK (Δ*dnaK*) deletions[15,16] suggests functional redundancy in the chaperone network. To study its molecular base, we performed SeRP in Δ*tig* and Δ*dnaK* mutants. TF enrichment profiles are not affected by *dnaK* deletion (Fig. 5a, b; Supplementary Fig. 5a, b), even for the few nascent proteins where DnaK engages before TF (e.g., AspC and DeoC, Supplementary Fig. 5c). In contrast, DnaK and GroEL showed stronger co-translational enrichment in Δ*tig* cells, evidenced by the elevated amounts of RNCs co-purified with DnaK and GroEL (Supplementary Fig. 5d). Two effects account for the stronger DnaK and GroEL enrichment. First, both chaperones engage additional substrates when TF is absent (Supplementary Fig. 5e), and second, while they do not precisely mimic the TF behavior, the established substrates are engaged earlier in translation (Fig. 5a, b; Supplementary Fig. 5a, b). These changes suggest altered chaperone competition for nascent chain binding. Supporting this notion, earlier DnaK engagement is limited to TF substrates (CP, PP, and OMPs) and is not observed for IMPs (Fig. 5a right). Further, the overall engagement shift is greater for nascent chains with high TF enrichment (Supplementary Fig. 5f). Performing GroEL-SeRP in Δ*dnaK* mutants did not provide evidence for competition between DnaK and GroEL (Fig. 5a, b; Supplementary Fig. 5g). Instead, we detected slightly delayed and weakened GroEL enrichment in the absence of DnaK (Supplementary Fig. 5g, h). While the effect size is small and the changes are at the edge of significance, this observation raises the interesting possibility that DnaK supports GroEL engagement with some nascent proteins.

## Chaperone action in the folding of inner membrane proteins

Our ribosome profiling data provide not only information on chaperone engagement with cytosolic proteins but also with proteins targeted to the translocon for translocation or inner membrane insertion. Here we focus on inner membrane protein (IMPs) as they often have cytosolic segments, which may be accessible for cytosolic chaperone action. While nascent IMPs generally do not bind TF, many IMPs (123)

are found within the cluster A of DnaK (Supplementary Fig. 6a). Strikingly, DnaK engages upon emergence of large cytoplasmic segments, consistent with DnaK assisting their folding (Fig. 6a, Supplementary Fig. 6b, c left). DnaK binding may be facilitated by the dissociation of ribosomes from the translocon before translation is completed. Histidine kinases are a prominent class of membrane receptors that sense environmental signals to regulate gene expression[61]. DnaK engages 14 out of 16 expressed histidine kinases co-translationally, generally upon exposure of C-terminal cytoplasmic domains implicated in phosphorylation (for example BarA, Fig. 6b). Another example is the DNA partitioning protein FtsK, to which DnaK binds when three C-terminal cytoplasmic domains emerge after a large unstructured region is translated (Supplementary Fig. 6d). This implies a role of DnaK in the folding of long cytoplasmic domains of IMPs that is reminiscent to its role in the folding cytoplasmic proteins (Fig. 6c).

The enrichment periods of GroEL are not limited to the emergence of cytosolic domains, suggesting an alternative role of GroEL (Supplementary Fig. 6c, e). We speculate that GroEL binds nascent IMPs that failed to be integrated into the membrane, to prevent aggregation and facilitate post-translational membrane insertion, similar to the support of post-translational membrane insertion of the IMP LacY in vitro[62]. The low amount of GroEL found in association with ribosomes[12] is in line with such a rescuing function.

## Discussion

This work reveals the co-translational mode of action of the ensemble of cytosolic chaperones involved in the folding of actively synthesized proteins in *E. coli*. We show that the nascent chain enrichment profiles of TF and DnaK are predictable for a major fraction of the proteome, based on the simple yet general notion of unsatisfied residues on partially folded structures, which we introduce here. Our findings contrast with long-held views that both TF and DnaK interact predominantly with unfolded substrate conformers[7,20,63], and rather indicate they bind more stably to partially folded nascent structures[6,64,65]

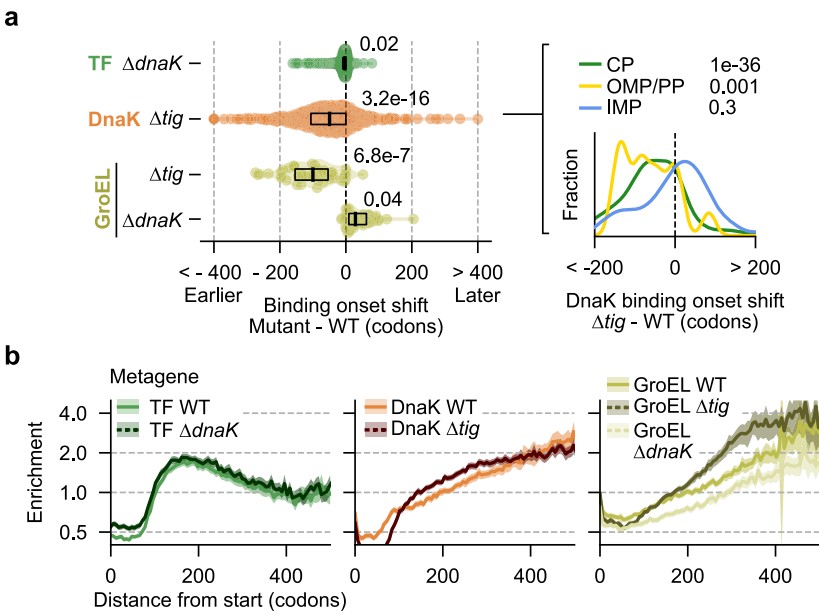

**Fig. 5 | Robustness of the chaperone network. a** Distribution of altered chaperone engagement onsets for high-affinity substrates upon genetic deletion of TF (Δ*tig*) or DnaK (Δ*dnaK*). Black bars indicate medians, and boxes indicate the interquartile range. Numbers indicate two-sided Kolmogorov–Smirnov test-derived *p* values comparing mutant and WT onsets (left). Altered DnaK onsets upon TF deletion sorted by localization: cytoplasm (CP, green, *n* = 389), translocated (OMP/PP,

yellow, *n* = 16), and inner membrane (IMP, blue, *n* = 81). *P*-values indicate the difference from 0 using a one-sided *T* test of the mean. (right). **b** Genome-wide (*n* = 3961) metagene enrichment profiles in different strain backgrounds. Solid lines and shadows indicate means and 95% CI, respectively. Source data are provided as a Source Data file.

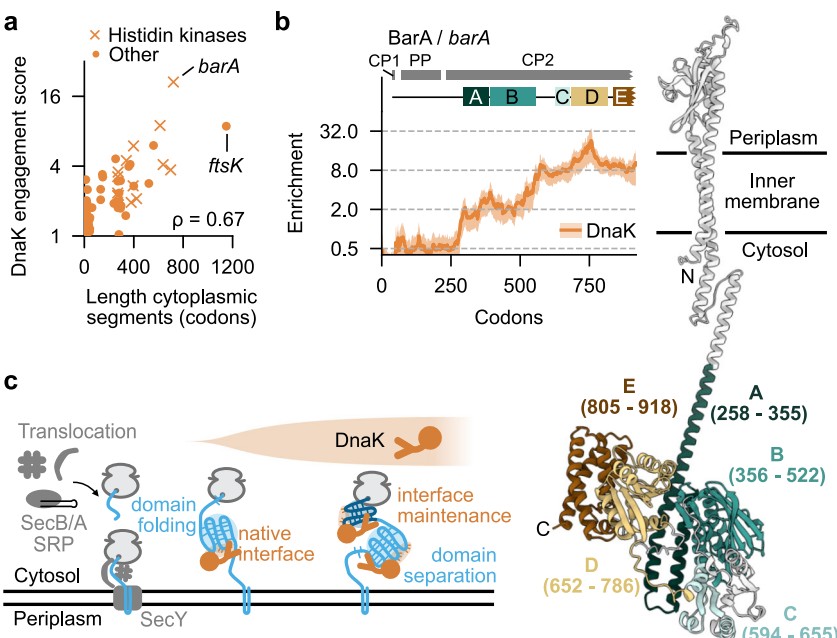

**Fig. 6 | Chaperone action on nascent inner membrane proteins. a** Scatterplot comparing the length of cytoplasmic segments (*x* axis) with the maximal DnaK engagement scores within these segments (*y* axis) for strongly engaged IMPs (*n* = 72). *ρ* indicates the Spearman correlation coefficient. **b** DnaK engagement profiles with nascent BarA (left). Grey bars indicate membrane topology of loop segments (CP cytoplasmic, PP periplasmic). Solid orange line indicates the average, shadow indicates the 95% CI. AlphaFold prediction of the BarA structure (AF-P0AEC5, right). **c** Schematic of co-translational folding support for multi-domain IMPs by DnaK. Source data are provided as a Source Data file.

across the proteome. Our models predict detailed features like chaperone engagement onset and dissociation during translation for proteins across the proteome, based only on protein sequence and general chaperone binding rules. These predictions matched the SeRP data for a large fraction of the cytosolic proteins. Such predictive ability is remarkable given the vast structural diversity of proteins and the dynamics and complexity of the underlying translation, co-translational folding, and chaperone guidance processes. At the same time, a general mechanism makes sense given the proteome-wide nature of chaperone engagement. We stress that, as is common for most models, our predictions should be taken as hypotheses. Indeed, the model's purpose was to gain a mechanistic understanding: whether simple rules that do not contain protein-specific chaperone binding information have predictive value for cross-proteome TF and DnaK binding. Note that our model thus does not aim to include all mechanisms, but rather to capture the proteome-wide common denominator for cytosolic proteins. For instance, we find that TF does not bind the more C-terminal domains in large multi-domain proteins, likely due to steric occlusion of the TF ribosomal binding site that is not included in the model.

Unsatisfied residues and their roles can be understood intuitively. They are residues that cannot form all native contacts because their partner residues are not yet all translated—and hence they can contact chaperones instead. Folding brings multiple unsatisfied residues close together on the resulting structure. Their proximity is closer and their configuration more rigid than for unfolded chain conformers, allowing more stable cooperative binding for both TF and DnaK and limiting the entropy loss upon chaperone binding.

Upon the start of translation, TF binding is promoted by (i) TF docking to ribosomal protein L23 near the tunnel exit, which increases the local concentration of TF and on-rate of nascent chain binding, thereby prioritizing TF over DnaK and confining TF´s action to the tunnel exit, and (ii) the growing number of unsatisfied residues exposed on partially folded structures that contact several sites in the inner surface of TF[17,18,20,30,66]. TF can likely also sample nascent molten

globule states prior to folding, but in a dynamic and multivalent manner that is less stable. While the emergence of new residues is determined by ongoing translation, folding is a stochastic process that can vary between the same nascent chain species. As a result, chaperone binding onsets can average out and become more gradual in the SeRP data. We find that TF binding generally correlates with intra-domain contacts, which can explain how TF protects partially folded domains from distant non-native interactions that cause misfolding[65]. While translation continues and domain folding is completed, intra-domain contacts become satisfied. TF thus reduces its affinity and dissociates, and paves the way for DnaK, which binds when unsatisfied residues involved in inter-domain contacts accumulate, likely using its binding groove to bind short unfolded nascent chain segments and its lid to bind partially folded structures[64,67,68]. Unlike TF, DnaK does not rely on direct ribosome binding, and thus can remain bound to N-terminal domains as they are pushed away from the ribosome by translation. DnaK enrichment typically does not decrease during translation, also suggesting that DnaK keeps domains separated until after translation[60,69,70], which should be particularly beneficial when non-neighboring domains must ultimately dock onto each other.

The chaperone network exhibits significant flexibility. Upon TF deletion, DnaK and GroEL bind earlier during translation and to more substrates—including the smaller proteins that are normally bound by TF. This response of DnaK and GroEL contrasts with TF, whose substrate pool and enrichment profiles are not affected by DnaK deletion. This adaptive triaging provides robustness to the chaperone network, and explains compensation of chaperone deficiencies in mutant cells[8,15,16,71,72].

We speculate that unsatisfied residues exposed on partially folded structures may be more broadly relevant to chaperone engagement, given their generic presence on folding intermediates and the ubiquity of co-translational folding across all domains of life. Whether and how other chaperones target unsatisfied residues remains to be studied. As it requires protein sequence only, our model offers hypotheses for artificially designed proteins[73]. Unsatisfied residues may also be

relevant to understanding post-translational chaperone guidance, ranging from Hsp70-mediated disaggregation[74] to Hsp70- and Hsp90-mediated kinase regulation[75], and may help in understanding the breakdown of the protein quality control machinery under stress[1]. It will also be important to study how the mechanisms observed here play a role in recently observed co-translational assembly processes[76–78].

# Methods

## Bacterial strains and plasmids

To explore the compaction of nascent chains, we purified RNCs displaying TrpD fragments of varying lengths. These lengths reflect snapshots of nascent chain folding states when TrpD N-terminal domain is partially exposed (TrpD[172]) or fully exposed (TrpD[192]). We constructed a series of plasmids encoding N-terminal Strep-tagged Sumo fused to N-terminal fragments of TrpD, followed by a linker and the minimal SecM stalling sequence, as previously described by ref. 66. Stalled ribosome nascent chain complexes were generated in BL21 Rosetta Δ*tig* cells, purified using Strep-tag AP, and the N-terminal Strep-tagged Sumo fusion was subsequently cleaved with a catalytically active fragment of the Sumo protease Ulp1[66].

All *aceE*-encoding plasmids used in optical tweezer experiments were derived from the pRSET-DHRF plasmid[38]. The DNA constructs encoding the different protein fragments were preceded by an N-terminal pre-sequence consisting of an amber stop codon and succeeded by a C-terminal strong SecM arrest peptide (FSTPVWIWWW-PRIRGPP). The different fragments of the *aceE* were combined with pRSET-DHRF, which was linearized by *Xho*I and *Nco*I. See Supplementary Table 1 for used primer and reagents.

## Purification of chaperone-RNC complexes and deep sequencing library preparation in *E. coli*

A fresh overnight culture of Δ*tig*::*tig*-TEV-Avi cells[32] was cultivated in EZ Rich defined medium (MOPS EZ Rich defined medium) supplemented with 40 μg/ml of D-biotin at 30 °C under rigorous shaking (110 rpm) to an OD$_{600}$ of 0.4 to 0.5. Then cells were harvested by rapid filtration and lysed by mixer milling (2 min, 30 Hz, MM400 Retsch) with frozen lysis buffer (50 mM HEPES-KOH pH 7.0, 100 mM NaCl, 10 mM MgCl$_2$, 1 mM Chloramphenicol, 1 mM PMSF, 0,4% Triton, 0.1% NP-40, 5 mM CaCl$_2$, 0.2% glucose, 1x protease inhibitors (Complete EDTA-free, Roche), 1x Apo, 1x Leupeptin and 1x PepA and RNase-free DNase I 0,1 U/μl. The pulverized cells were thawed in a 25 °C water bath for 1 min and incubated for 10 min in an ice-water bath. The purification of TF-RNCs complex were prepared according to ref. 32 skipping the chemical cross-linking reaction.

For the purification of the DnaK-RNCs, lysates were thawed by stepwise addition of cold 100 μl hexokinase buffer (50 mM HEPES-KOH pH 7.0, 40 mM sodium phosphate buffer pH 7.0, 0.2% glucose 100 U hexokinase) while stirring with a magnetic flea to immediately deplete ATP. ATP depletion was used before to maintain Hsp70-nascent chain interactions in SeRP[79,80]. All lysates were cleared by centrifugation for 5 min at 20,000 × *g*, 4 °C. Supernatants were digested using 150 U MNase per 1 A$_{260}$ unit of RNA for 5 min at 25 °C. Digest was terminated by addition of 6 mM EGTA and chilling on ice. Digested lysates were loaded on sucrose cushions (25% sucrose, 50 mM HEPES-KOH pH 7.0, 500 mM NaCl, 10 mM MgCl$_2$, 1 mM Chloramphenicol, 1 mM PMSF, 40 mM sodium phosphate buffer pH 7.0) and centrifuged for 90 min at 75,000 rpm, 4 °C in a TLA120 rotor (250,000 × *g*). Ribosome pellets were resuspended in lysis buffer lacking CaCl$_2$. RNA concentration was determined by Bioanalyzer, and 100 μg total RNA was taken for the total translatome. For IP, 4 ml Dynabeads (Life Technologies) were washed three times with 4 ml TBS and incubated with 100 μl anti-DnaK antibody for 15 min at RT under constant shaking. Then, beads were washed three times with 4 ml of TBST buffer. Ribosomes were incubated with affinity beads for 30 min at 4 °C under

constant shaking. The matrix was quickly washed three times with cold wash buffer (TBS, 1 mM chloramphenicol, 10 mM MgCl2, 250 mM NaCl, 0.1% Triton X100). Phenol-chloroform extraction and deep sequencing library preparation were done as described in ref. 33. Sequencing was performed at the DKFZ Genomics & Proteomics Core facility for sequencing on a HiSeq 2000 from Illumina following the manufacture's recommendation.

For GroEL-RNC purification lysates were treated as for the DnaK IP. To obtain a quantity of RNA sufficient to prepare a cDNA library for sequencing, additionally, ATP-depleted 60 mM EDC (1-ethyl-3-[3-dimethylaminopropyl] carbodiimide; Pierce) crosslinker in lysis buffer was added to the thawed lysate and incubated for 5 min on an Eppendorf Thermomixer at 300 rpm at 16 °C. The cross-linking reaction was quenched with 200 mM Tris-HCl pH 8.0, 100 mM Glycine, and 4 mM NaHCO$_3$ and further incubated for 5 min on ice. Further steps were done as described for DnaK however using an anti-GroEL antibody. See Supplementary Table 1 for used reagents.

## Alignment and preprocessing of sequencing reads in *E. coli*

We first remove adapter sequences using cutadapt with the following parameter: "-u: 2, -nextseq-trim: 20, --discard-untrimmed, -m: 20 -M:45 -O:6 -a:NNNNNATCGTAGATCGGAAGAGCACACGTCTGAACTCCAG TCAC. Trimmed reads were first aligned against an rRNA/tRNA reference using bowtie1 with the following parameter: "-t, -n: 2, --best, --un: <outfile>". Unaligned reads from the previous step were aligned against the *E. coli* MC4100 reference genome using bowtie1 with the following parameter: "-t, -n: 2, -m: 1, --best, --strata". P-site positions were assigned using a fixed offset of 15 nucleotides from the 3'-end of reads. Further positions within coding sites were summed every three nucleotides to account for codons. We excluded four genes from analysis: The genes of the elongation factor *tufA* and *tufB* due to high sequence similarity and resulting alignment gaps; DnaK and GroEL because the IP-antibody also recognized nascent DnaK and GroEL, causing artificial enrichments.

## Chaperone engagement score analysis

Chaperone engagement scores were calculated as described in ref. 36. In short, for each transcript, a sliding window of 15 codons was applied. Then, the 95% confidence interval (CI) according to Agresti and Coull of the enrichment ratio (factor-bound translatome to total translatome) was calculated. This calculation was performed for each biological replicate separately, and then the average of both replicates was formed. For single-value engagement scores, we calculated position-wise CIs for each gene, but using a sliding window of 45 codons. We define the score as the highest value of the lower CI boundary, excluding the first 30 and the last 10 codons to avoid artifacts due to initiation or nearby downstream genes.

## Binding onset determination

Given a threshold enrichment value of 1.5, each position in a gene was assigned a numerical value of: 1 (bound with high confidence if the lower CI bound is above the threshold), 0.2 (bound with low confidence if the CI overlaps the threshold), 0 (not bound if the upper CI bound is below the threshold). To define enrichment peaks, two replicates were averaged, and position-wise binding values were thresholded at 0.6. Peaks shorter than 6 codons were discarded. The most 5' position above the threshold is defined as chaperone binding onset.

## Metagene analysis

If not stated otherwise, coding sequences were aligned to the start or stop codon, averaged between replicates and binned with a width of 6 codons. Then, the average enrichment at every position was calculated and normalized by the ratio between the AP or IP and total translatome library size. For domain wise metagene profiles, domain and linker

lengths were normalized to 30 and 10 bin, respectively. Linkers longer than 40 amino acids were excluded from the analysis.

## Cluster analysis

Low CI borders of every transcript were compared pair-wise using the dynamic time-warping module with standard parameters from the ts-learn package[81]. Genes shorter than 50 codons and the first 30 and last 10 codons of every transcript were ignored. The resulting similarity matrix was clipped for values larger than 500 (only relevant for *sdhA* and *frdA* in the GroEL IP) and then directly used for hierarchical clustering using the scipy.linkage module with the "ward" distance method.

## Domain annotation

Protein domains were obtained using the CATH sequence similarity search API and the FunFHMMer web server for all MC4100 protein sequences[82]. Domains from the resulting alignment longer than 10 codons were assigned after their similarity rank (highest ranked domains first). Lower-ranked domains were only aligned if the positional overlap with accepted higher-ranked domains was below or equal to 5%.

## Chaperone binding prediction

AlphaFold-derived mmcif files and predicted aligned error matrices for cytosolic *E. coli* proteins were obtained from https://alphafold.ebi.ac.uk/download (UP000000625_83333_ECOLI_v3) and via the urllib python package, respectively. Contacts within structure predictions are defined by atoms that are equal or closer than 6 Å to each other. Atoms of residues that have per a residue confidence score below 70 are ignored. Residue pairs that have a predicted aligned error larger than 5 Å are ignored. Corresponding residues that are equal or closer than 5 residues to each other on the peptide chain are ignored, which largely excludes residue contacts within individual α-helices. The number of atoms involved in a contacting residue pair was defined as the interaction strength. To gain codon resolution for chaperone binding predictions, we elongated the nascent chain by one residue or codon. To account for the ribosomal exit tunnel, we prohibited contact formation of residues that are up to 30 amino acids away from the attached residue (C-terminus). For every nascent chain length, we then sorted contacting residue pairs into three categories: I) both residues are prohibited from forming contacts or are not attached to the peptide chain at this stage of translation. II) The more N-terminal residue can form contacts, but the more C-terminal residue is prohibited from forming contacts or is not attached to the peptide chain. III) Both residues can form contacts. Contacts from category I are ignored. Contacts from category II were considered to contribute to chaperone binding, and the involved residues were named unsatisfied. Contacts from category III were considered to be part of compacted regions. For the "molten globule" model, every tunnel-emerged residue was considered to be engaged in category II contacts until all residues of a domain were available, which satisfied all residues immediately. For the "delayed folding" model, we excluded category II contacts from flexible C-terminal regions. Therefore, we calculated the cumulative sum of category III contact interaction strengths from the C- to N-terminus, ignoring residues with an individual contact strength below 20. We only considered category II contacts for chaperone binding in those N-terminal regions where the cumulative category III contact interaction strengths were larger than 750. Such category II contacts were then sorted after their CATH domain annotation: Unsatisfied contacts where both residues are within an annotated domain were used to predict TF interaction, whereas unsatisfied contact residues that are split between annotated domains were used to predict DnaK interaction. For the DnaK model, category III contacts, which are between domains, are always considered as category II, which accounts for the separation of protein domains until translation termination. The codon-specific chaperone binding value was then calculated by the sum of the category II contact interaction strengths. For single- and metagene plots, the model output was normalized between 0.25 and the SeRP derived chaperone engagement score (max low CI) of the predicted gene to account for differences in gene expression levels and chaperone background binding.

## Limited proteolysis of TrpD RNCs

To investigate the length-dependent nascent chain folding status purified stalled RNCs exposing two different TrpD nascent chain lengths (TrpD[172] and TrpD[192]) were probed for trypsin sensitivity. 2 μg of stalled RNCs were incubated with varying trypsin concentrations (0.25 dilution series starting final concentration 1.2 μg/ml) for 15 min at 25 °C. Proteolytic digest was stopped by addition of PMSF protease inhibitor (final concentration 2 mM), directly followed by NaOH treatment (final concentration 0.1 M) to release P-site tRNA from the nascent chain. After incubation on ice for 7 min 5x SDS sample buffer was added followed by incubation at 95 °C for 5 min. Proteolytic cleavage patterns were analyzed by Tricine-SDS-PAGE[83] and immunoblotting using α-SecM primary antibody.

## Optical tweezers: neutravidin-DNA handle preparation and coupling to beads

Double-stranded DNA (dsDNA) handles were generated via PCR amplification, utilizing digoxigenin (DIG) and biotin 5′-end-labeled primers. The primers 2DIGfw5kbp and 3BIOrev5kbp were paired with the pOSIP-TT template in a Phire Green Hot Start II PCR (Thermo Scientific). The resulting 5 kb PCR output was refined using the QIAquick PCR Purification Kit from QIAGEN. This 5 kb DNA, at a concentration of 10 nM, was then incubated with 2 μM neutravidin (Thermo Scientific) in 10 mM PBS at 4 °C overnight. Anti-DIG beads were prepared by cross-linking anti-DIG (Roche) to carboxyl-functionalized polystyrene beads of 2.1 μm size, sourced from Spherotech, using the EDAC carbodiimide crosslinker (PolyLink protein coupling kit by Polysciences). Following this, the 5 kbp neutravidin-DNA strands, having a contour length of 1.7 μm, were attached to the anti-DIG beads, maintaining a ratio of approximately 10 neutravidin-DNA per bead, and left for 30 minutes at 4 °C. After coupling, these beads, now with neutravidin-DNA attachments, underwent multiple washes with Tico buffer, which consists of 20 mM HEPES-KOH pH 7.6, 10 mM (Ac)2Mg, 30 mM AcNH4, and 4 mM β-mercaptoethanol. The final product was then divided into two separate low-protein-binding tubes.

## Optical tweezers: coupling ribosomes to beads

Ribosomes sourced from an *E. coli* K-12 strain (Can20/12E3) that lacks RNase were avi-tagged in vivo at the uL4 ribosomal protein, subsequently biotinylated, and were later extracted. These biotin-tagged ribosomes were then added to a set of neutravidin-DNA modified beads (that had undergone a Tico buffer wash) at a concentration of 350 nM. RNase inhibitor (New England Biolabs) was added and left to incubate at 4 °C for a duration of 30 minutes. Post incubation, ribosomes that hadn't bound to the beads were separated out through a pelleting process. Following this, the beads underwent a single wash using Tico buffer and were subsequently immersed directly into a cell-free transcription/translation concoction as detailed further.

## Optical tweezers: cell-free protein synthesis with TF and DnaK and co-translational labeling of the nascent chain

For cell-free protein synthesis, a modified reconstituted bacterial PURE system devoid of ribosomes (PUREexpress Δ ribosomes, New England Biolabs) was employed. Co-translational incorporation of biotin at the N-terminal amber sites (TAG) was achieved via the suppressor tRNA methodology, as described in ref. 38. This system was supplemented

with 10 µM of a tRNA pre-loaded with biotinylated lysine (Biotin-XX-AF_tRNA, amber, CloverDirect), and murine RNase inhibitor sourced from New England Biolabs. 100 nM of fluorescently labeled TF and DnaK, with or without 17 nM DnaJ and 10 nM of GrpE, were added (Supplementary Fig. 3a). Then, after mixing the previously prepared bead-coupled ribosomes to this partially assembled reaction mix, the initiation of synthesis was triggered by adding a linear DNA template of one of the AceE fragments (see section "Bacterial Strains and Plasmids" above) at a concentration of 5.5 nM. This reaction was then kept at 37 °C for 30 minutes. Owing to the SecMstr arrest peptide located at the C-terminus, the emergent nascent chains remained anchored to the ribosome[84]. Post the transcription/translation phase, the beads, now bound with halted RNCs, were reconstituted in pre-cooled TICO buffer (20 mM HEPES-KOH pH 7.5, 10 mM (Ac)2Mg, 30 mM AcNH4, 4 mM β-mercaptoethanol) and maintained at 4 °C. For optical tweezer applications, these RNC-bound beads were diluted in a 300 µL volume of TICO buffer. The experiments were performed in the C-Trap in a TICO buffer containing 100 nM of Atto 532 and Atto 647 N-labeled TF & DnaK, with or without 17 nM DnaJ and 10 nM of GrpE, and 1 mM ATP & regeneration system (20 ng/µl pyruvate kinase, 3 mM phosphoenolpyruvate). 1 mM Trolox in combination with the P2O system, comprising 3 units/ml pyranose oxidase, 90 units/ml catalase, and 50 mM glucose (all from Sigma), served as the oxygen scavenger.

## Optical tweezers: chaperone labelling

For single-molecule optical tweezer experiments combined with fluorescence microscopy, translation of *aceE-secM* was performed in the presence of 100 nM ATTO532 labelled DnaK as well as 100 nM ATTO647N labelled TF. The chaperones DnaK-290C[85] and TF-99C[86] were dialyzed overnight at 4 °C in PBS buffer supplemented with 1 mM Tris(2-carboxyetyl) phosphine hydrochloride (TCEP). Subsequently, protein concentrations ranging from 2 to 10 mg/ml were dissolved in a 5 mM TCEP solution. Fluorescent dyes were solubilized in N, N-Dimethylformamide and labeling was started by mixing the chaperones with an excess of the dyes of about 20-fold (for DnaK) and 10-fold for TF. This reaction mixture was gently stirred and protected from light for 2 hours at 25 °C, and the reaction was stopped by the addition of 10 mM DTT. The reaction was kept for 2 hours at 25 °C while gently stirring and protecting the vial from light, and stopped by adding 10 mM DTT. Any excess label was removed by size exclusion chromatography using a Sephadex 75 10/300 GL column.

## Optical tweezers & correlated confocal microscopy assay and single-molecule data analysis

All single-molecule measurements were performed using the C-Trap instrument (LUMICKS, Amsterdam), with two-color confocal fluorescence (532 nm and 638 nm) and APDs for single-photon sensitivity. Two optical traps are formed from a single 1064 nm laser, and can be steered relative to each other with high precision using a piezo mirror. The monolithic laminar flow cell incorporated in the instrument supports a passive pressure-driven microfluidic system with five distinct flow channels. Data collection occurred at a frequency of 50 kHz, averaged to 500 Hz, and saved at that frequency, and subsequent analysis was conducted using tailored scripts in both Matlab and Python. A cycling force spectroscopy mode was employed for measurements. Herein, one of the traps was moved at a constant speed, ranging between a minimal bead separation of 2 µm and a maximum force reaching 65 pN. Simultaneously, both fluorescence excitation beams (532 nm and 638 nm) were scanned along the tethered molecule at a rate of 10 Hz. The tethered molecule was either one of the AceE nascent chain fragments. Binding events of TF and DnaK for AceE fragments were recorded as a single fluorescent spot appearing between both optically trapped beads (Fig. 3b), and this fluorescence signal in time was aligned to the force signal (Supplementary Fig. 3c).

The assay is based on previous work[35,38–40] that included controls for: (1) Nascent chain synthesis[38]. Bulk PURE in vitro transcription-translation (IVTT) reaction translating a GFP construct without Histidine did not show GFP fluorescence, while adding Histidine did yield measured GFP fluorescence. (2) Nascent RNC tethering. DNA handles with RNCs in between show larger nm-scale length fluctuations than DNA handles only[87]. (3) DNA handle attachment to biotinylated nascent chains. For ribosomes not exposing a nascent chain by omitting the translation reaction, correct tethers showing the right length and unfolding events were not obtained (no correct tethers in 53 bead-approach attempts), while correct tethers were reliably obtained when the ribosomes did expose a biotinylated nascent chain (23 correct tethers in 68 bead-approach attempts). (4) Nascent chain attachment and (un)folding[39]. Ribosomes were tethered that had translated a polypeptide segment that either did contain, or as a control did not contain, the protein of interest. The former showed unfolding and refolding events of the length expected for the protein, whereas the latter did not show any unfolding or refolding features. (5) Fluorescent chaperone binding to tethered polypeptide chain[35,40]. The presence of Atto647N-labeled TF in solution resulted in transient fluorescent spots as seen in this study, at the location of the tethered polypeptide chain (in between beads), whereas the absence of labelled TF did not show such spots. (6) Fluorescent chaperones bind specifically to nascent. Our measurements presented here (Fig. 3) show that both TF and DnaK do not bind when the nascent chain is short and do bind when the nascent chain is longer, as expected. If the chaperones had bound to components other than the nascent chain, one would have detected binding also in the shortest construct.

For Fig. 3d, e, Supplementary Fig. 3d, we analyzed those experiments for which the level of compaction could be determined, which, for instance, depended on whether the nascent chain could be fully unfolded. In order to determine the level of compaction of nascent chains and to ensure single tethers were formed between the optically trapped beads, the resulting force-extension curves for each tether was fitted with two sequential worm-like chains: one accounting for the DNA handle (extensible worm-like chain eWLC) and the other for the stalled nascent chain (inextensible worm-like chain WLC). For molecules that are fully unfolded, the state of nascent chain compaction could be determined at the point of chaperone binding by fitting an eWLC ruler. Partially folded states with a measured contour length (the length of the unfolded part of the chain) that was greater than half of contour length of the fully unfolded state (for that particular AceE fragment) were categorized as ">50% unfolded". Partially folded states where the measured contour length (length of the unfolded part of the chain) was less than half of the contour length of the fully unfolded state were categorized as ">50% folded". By repeatedly waiting at zero force, followed by stretching and relaxing, the nascent chain had a chance to repeatedly cycle through several different states of compaction.

## Statistics and reproducibility

If not stated otherwise, the mean of two biological replicates is shown as solid lines. Shadows indicate the 95% CI. For violin plots, black bars indicate medians, and boxes indicate the interquartile range. If not stated otherwise, the two-sided Mann–Whitney $U$ test was applied to calculate $p$ values. The following python packages were used: numpy for mean and median calculations, scipy for clustering, statistical tests, and regression analysis. statsmodels for CI calculation according to Agresti and Coull.

For Fig. 3c, bottom, a total of 796 molecules (all AceE fragments) were measured as described above, resulting in a total of 173 TF, and 162 DnaK binding events. One of the 173 TF binding events is shown in Fig. 3b. Fragment 1 featured no chaperone binding ($n = 54$ molecules), fragment 2 featured 58 TF binding events and no DnaK binding events ($n = 115$ molecules), fragment 3 featured 65 TF and 37 DnaK binding

events ($n = 226$ molecules), fragment 4 featured 47 TF and 64 DnaK binding events ($n = 209$ molecules), fragment 5 featured 3 TF binding and 51 DnaK binding events ($n = 97$ molecules), and fragment 6 featured 0 TF and 10 DnaK binding events ($n = 95$ molecules). Figure 3c, bottom compares the occupancy of TF and DnaK between AceE fragments. The displayed occupancy values are relative, and add up to unity for each chaperone. To determine these values, we first added up the number of TF binding events of each fragment and divided by the total number of molecules measured for each fragment. To account for different measurement times, we divided the resulting fractions by the total measurement time of each fragment. To obtain normalized values, we divided these resulting fractions by the sum of all the fractions. The DnaK values were determined in the same way. Error bars show the standard error of proportion calculated by the following formula (1):

$$\text{Standard error per fragment n} = \sqrt{\frac{p_n(1 - p_n)}{N_n}};$$

where $p$ is the sample proportion given by the calculated normalized fractions above. $N$ is the sum of all the molecules for each fragment.

Figure 3e features $n = 40$ TF binding events (majority folded: $n = 21$, majority unfolded: $n = 19$) and $n = 33$ DnaK binding events (majority folded: $n = 12$, majority unfolded: $n = 21$) for which the level of compaction could be determined as described above. Whilst Fig. 3e shows binding times for all AceE fragments and distinguishes between majority folded and unfolded, Fig. 3d displays individual binding times and corresponding chain compactions for AceE fragment 2 for TF binding events ($n = 33$) and AceE fragment 5 for DnaK binding events ($n = 21$). The correlation coefficients of Fig. 3d were calculated with Spearman's correlation to test for monotonic relationship rather than solely linear relationships. Supplementary Fig. 3d features the following number of TF binding events for AceE: 27 (fragment 2), 15 (fragment 3), 18 (fragment 4), 0 (fragment 5), 0 (fragment 6), and the following DnaK binding events for AceE: 0 (fragment 2), 6 (fragment 3), 10 (fragment 4), 21 (fragment 5), 4 (fragment 6).

### Reporting summary
Further information on research design is available in the Nature Portfolio Reporting Summary linked to this article.

## Data availability
All data necessary to interpret, verify, and extend the research (processed ribosome profiling data and chaperone enrichment CIs) in this article have been deposited at the open science foundation (OSF) [https://doi.org/10.17605/OSF.IO/SH4YQ]. Raw sequencing data are accessible at the Gene Expression Omnibus repository with the accession code GSE292386. Data is publicly available as of the date of publication. Unless otherwise stated, all data supporting the results of this study can be found in the article, supplementary, source data files or the OSF website. Source data are provided with this study. Source data are provided with this paper.

## Code availability
All relevant custom codes used in this work have been deposited at the OSF [https://doi.org/10.17605/OSF.IO/SH4YQ] or zenodo/github [https://doi.org/10.5281/zenodo.15025183]. Code is publicly available as of the date of publication.

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

## Acknowledgements

B.B. and S.J.T. acknowledge a research grant of the European Union (ERC - SyG - 101072047 - CoTransComplex). Views and opinions expressed are, however, those of the authors only and do not necessarily reflect those of the European Union or the European Research Council. Neither the European Union nor the granting authority can be held responsible for them. This work was also supported by a research grant of the Deutsche Forschungsgemeinschaft to GK (KR 3593/2-1). C.V.G. was supported by the European Commission (Horizon 2020, Marie Skłodowska-Curie Individual Fellowship, RiboEscapers–101024158). E.P.O. acknowledges support from the National Science Foundation (MCB 2031584) and National Institutes of Health (R35-GM124818). F.W. received funding from the European Union's Horizon 2020 Research and Innovation Programme under the Marie Skłodowska-Curie grant agreement No. 745798. We thank Alexandros Katranidis for providing biotinylated Ribosomes, and Bibiane Jager for help with control experiments. Work in the lab of S.J.T. was supported by the Netherlands Organization for Scientific Research (NWO). We thank the Klaus Tschira Foundation for its generous support in the acquisition of the sequencing device.

## Author contributions

B.B., G.K., and S.T. conceived the study. C.V.G., G.K., B.B., F.W., and S.T. designed the experiments. C.V.G., F.W., and J.J.A. conducted the experiments. F.T., I.K., M.G., K.T., C.V.G., and E.O. analyzed the data. All authors contributed to the writing and critical evaluation of the manuscript.

## Funding

## Competing interests

The authors declare no competing interests.
