## [Transparent Peer Review file · Nature Communications]

Proteome-wide determinants of co-translational chaperone binding in bacteria

Corresponding Author: Professor Bernd Bukau

This manuscript has been previously reviewed at another journal. This document only contains information relating to versions considered at Nature Communications. Mentions of the other journal have been redacted.

Version 0:

Reviewer comments:

Reviewer #1

(Remarks to the Author)

The manuscript as revised for this submission satisfactorily addresses the review that I submitted on the past version, which was submitted to **[Redacted]**.

(Remarks on code availability)

Reviewer #5

(Remarks to the Author)

My assessment regards the authors' responses to Reviewer 3.

In the latest version of the paper, the authors' tone-down regarding the generality of the model is appreciated. Furthermore, the section about the model, with the many additional details, now covers all the critical points in the description of the approach. The same applies to the clarifications regarding the experimental procedures and the references to their previous development and applications. The removal of the paragraph on yeast makes the presentation of the content and results clear and homogeneous.

Overall, thanks to the applied methodologies, the presented data, and the quality of the text and figures, I believe that the paper in its current form meets the standards required for publication in Nature Communications.

(Remarks on code availability)

I have not had the opportunity to install and test the code. However, I have inspected the items uploaded at OSF and believe that the scripts are well-documented, with a README that comprehensively lists the available scripts and the data to be used for the demo. A good option would be to create a similar repository on GitHub which, thanks to its growing popularity, could help disseminate the authors' computational contribution.
